# Size- and Chirality-Dependent Structural and Mechanical Properties of Single-Walled Phenine Nanotubes

**DOI:** 10.3390/ma16134706

**Published:** 2023-06-29

**Authors:** Yanjun Liu, Ruijie Wang, Liya Wang, Jun Xia, Chengyuan Wang, Chun Tang

**Affiliations:** 1Faculty of Civil Engineering and Mechanics, Jiangsu University, Zhenjiang 212013, China; 2Zienkiewicz Centre for Computational Engineering, Faculty of Science and Engineering, Bay Campus, Swansea University, Swansea SA1 8EN, Wales, UK; 3Key Laboratory for Intelligent Nano Materials and Devices of Ministry of Education, Nanjing University of Aeronautics and Astronautics, Nanjing 210016, China

**Keywords:** phenine nanotubes, mechanical properties, molecular dynamics, bond twisting

## Abstract

Phenine nanotubes (PNTs) have recently been synthesized as a promising new one-dimensional material for high-performance electronics. The periodically distributed vacancy defects in PNTs result in novel semiconducting properties, but may also compromise their mechanical properties. However, the role of these defects in modifying the structural and mechanical properties is not yet well understood. To address this, we conducted systematic molecular dynamics simulations investigating the structural evolution and mechanical responses of PNTs under various conditions. Our results demonstrated that the twisting of linear carbon chains in both armchair and zigzag PNTs led to interesting structural transitions, which were sensitive to chiralities and diameters. Additionally, when subjected to tensile and compressive loading, PNTs’ cross-sectional geometry and untwisting of linear carbon chains resulted in distinct mechanical properties compared to carbon nanotubes. Our findings provide comprehensive insights into the fundamental properties of these new structures while uncovering a new mechanism for modifying the mechanical properties of one-dimensional nanostructures through the twisting–untwisting of linear carbon chains.

## 1. Introduction

One-dimensional (1D) tubular structures have garnered considerable interest from the scientific community due to their remarkable features that make them useful in various applications, including the reinforcement of composites [1,2,3,4], electronic devices [5,6,7,8], drug delivery [2,9], among others. Specifically, carbon nanotubes (CNTs) possess unique properties such as a narrow and seamless crystal structure, exceptional electronic transport properties, and chemical stability [1,10,11,12,13,14,15,16], which make them promising candidates for electronic devices. However, fully exploiting the potential of CNTs for the semiconducting industry remains uncertain, largely due to their band gap being dependent on their chiralities. Therefore, synthesized CNT arrays often consist of both metallic and semiconducting tubes. While some strategies have been developed to address this issue, such as breaking down metallic CNTs [17,18,19,20,21] or synthesizing CNTs with specific chiralities [22,23,24,25,26], these methods usually result in low yields and are therefore unsuitable for industrialization. Recently, Sun et al. succeeded in synthesizing a novel type of tubular carbon called phenine nanotubes (PNTs). PNTs can be thought of as porous graphene layers rolled into tubes [27]. One advantage of this member of the carbon allotrope family is that it possesses a finite band gap that is well-suited for semiconducting applications.

While the promising electronic properties of PNTs have been well studied, less is known about their mechanical properties in comparison to perfect CNTs or those with localized defects. It is widely accepted that the presence of defects in PNTs will lead to a deterioration in their ability to withstand external loads, but the unique porous nature of this new structure and its impact on the structural, elastic, and failure behaviors of PNTs remains unclear. CNTs are known for their extremely high strength and can undergo plastic behavior at high temperatures [3,11,28,29,30] and through various types of defect formation such as Stone–Wales transformation. However, vacancy defects typically reduce the strength of CNTs and do not facilitate ductile deformation [31,32,33]. PNTs with regularly patterned pores can be considered as a specific type of CNTs with periodically arranged vacancies, suggesting that the mechanical strength of PNTs is likely to be degraded; this is confirmed by recent simulation results [34]. It is important to note that defects in PNTs are not localized, resulting in a weakened tubular skeleton along the transverse direction. Recent calculations by Yu et al. [35] showed that the cross-sectional geometry of PNTs does not retain a circular shape for some cases, thereby causing the collapse of bundles of small-sized PNTs to flattened structures. However, there has been no systematic investigation into how these geometric changes affect the mechanical properties of PNTs. Such features are likely correlated to the sizes and chiralities of the PNTs and require comprehensive studies into their atomistic details.

In this study, we have examined the structural and mechanical properties of armchair and zigzag periodic nanotubes (PNTs) using molecular dynamics simulations. Our results have shown that PNTs exhibit unique structural features based on their diameter and chirality. For armchair PNTs, the cross-sectional shape changes from a circular to a polygonal shape as the diameter increases. When the diameter is greater than 50 Å, complex structures are observed. On the other hand, for zigzag PNTs, the circular cross-sectional shape remains until the diameter reaches around 50 Å, beyond which it transforms into an oval shape. At diameters larger than 150 Å, it tends to collapse towards a flattened geometry. Such anisotropic structural features lead to distinct mechanical properties of PNTs under tensile and compressive loads, which are common loads in mechanical devices. We have investigated the elastic properties and failure modes of PNTs, which were found to show size dependency. Our study provides fundamental insights into how periodic defects modify the structural and mechanical properties of 1D nanotube systems. This allows us to establish a comprehensive database for further studies when selecting this novel structure for functional nano devices.

## 2. Simulation Details

We conducted all simulations using the large-scale atomic/molecular massively parallel simulator (LAMMPS) package [36]. Visualization was carried out using the OVITO package [37]. The construction of the PNT structures is described in Section 3. To describe interatomic interactions, we adopted the AIREBO potential [38], which has proven successful in studying the structural and mechanical properties of carbon nanomaterials. To avoid a nonphysical description of mechanical properties by the original AIREBO potential, we modified the cutoff distance to 2 Å [39,40,41]. Geometry optimization was first performed using the conjugate gradient method with periodic boundary conditions. The system was then relaxed in the NVT ensemble with a time step of 1 fs until it reached an equilibrium state; this time step is commonly used in studying nano carbon systems [34,42]. It is worth noting that the exact values of the mechanical properties could be different if one choose other force fields, such as the reax force field. As the focus of this work is to uncover a correlation between structural evolution and mechanical responses, we did not concentrate our discussions on this aspect.

For investigating the mechanical properties of the structure, it was uniformly stretched or compressed along its axial direction with a strain rate of 0.1/ps until failure, which occurred at 1 K in the NVT ensemble. The results from this strain rate are comparable to those obtained by Yu et al. [35]. This allowed us to obtain insights into the behavior of the structure under tensile and compressive loading, and provided valuable information on its elastic properties and failure modes.

## 3. Results and Discussion

PNTs can be viewed as seamless rolled-up structures of graphene sheets, analogous to CNTs. As shown in Figure 1, rolling up along different directions leads to PNTs with different chiralities, such as armchair and zigzag types. For the ease of explanation, if the sheet is rolled up along x direction (corresponding to armchair edge) in the left panel of Figure 1, an armchair PNT is produced, as shown in the top right of Figure 1, while if rolled up vertically along the y direction (corresponding to zigzag edge), a zigzag PNT is obtained, as presented in the lower right panel of Figure 1. Due to the presence of defects, PNTs exhibited distinct structural properties compared to perfect CNTs. Our MD simulations revealed these differences for both armchair and zigzag PNTs, as seen in Figure 2, Figure 3 and Figure 4. To ensure equilibrium states were reached, energy minimization was performed first followed by relaxation at 1 K for 100 ps, or up to 2500 ps for larger models.

In the case of armchair PNTs shown in Figure 2, an interesting phenomenon was observed during energy minimization: the cross-sectional geometry transitions to a polygonal type when the diameter is small (less than 20.6 Å), while no sudden changes were observed during structural relaxation at a finite temperature for the duration of the simulations. This transition is attributed to the twisting of the linear carbon chain along zigzag edges, which reduces the PNT’s energy by about 0.33 eV/atom (Appendix A). The schematic twisting behavior is presented in Figure 3a, where a C-C bond linked to 2 benzene rings can be rotated along the bond direction; a series of rotations along the zigzag chain leads to notable reduction in energy. As a result, the type of polygon formed corresponds exactly to the number of linear zigzag chains in the PNT structure. As the diameter of the PNT increases, the trend varies, as depicted in Figure 2. For example, for a PNT with diameter D = 24.75 Å and 12 linear zigzag chains, the energy-minimized geometry exhibits a hexagonal cross-sectional structure that deforms further after finite-temperature relaxation. Consequently, the twisting angles of different linear carbon chains vary, indicating that the deformation modes become inhomogeneous and localized. The details of twisting angle variations can be found in Figure 3. With further diameter increases, the cross-section becomes more severely deformed and even collapses into a flattened shape at D = 53.6 Å. This effect arises because the linear carbon chain along the axis of the PNT is no longer able to resist radial deformation, and the van der Waals interaction between opposite walls plays a dominant role in reducing the total energy of the PNT system. From the potential energy plot and selected geometry evolution shown in Appendix A, it can be observed that for small-diametered PNTs, energy reduction during the relaxation process primarily comes from the twisting of carbon chains. On the other hand, for larger-diametered PNTs, the energy decreases gradually initially, suggesting that the contribution from twisting is less significant, while the flattening of the tube wall leads to a more significant decrease in potential energy (see, e.g., Appendix A), consistent with the above analysis.

The evolution of the cross sectional shape in zigzag PNTs displays a different trend, owing to the fact that there is an alignment of the carbon chains either circularly or as a 30° helix along the axis. The high axial symmetry resulting from these alignments prevents any major transformation to a polygon type. Nonetheless, the twisting of the zigzag carbon chains causes an increase in wall thickness, as explicated in Figure 4. In small diametered zigzag PNTs, the twisting angle is notably strong compared to that seen in armchair PNTs. Larger PNTs, however, do not exhibit evenly distributed twisting. Check Appendix A for further details. Due to the helical and circular alignment of carbon chains, PNTs show a slower collapse trend with an increasing diameter. It was not until D = 157.2 Å that the completely flattened structure was observed in zigzag PNTs, almost thrice that of their armchair counterparts. Nonetheless, there are certain overall similarities in the size-dependent correlation between the energy release mechanism and geometry revolution, as seen in Appendix A. Thus, we shall skip any detailed discussions of these shared premises here.

We next proceed to investigate the mechanical behavior of PNTs under various strain conditions. Figure 5 displays the tensile responses of selected armchair and zigzag PNTs. It is evident from the stress–strain relationship that both PNT types are brittle materials and undergo structural failure after reaching peak stress values, resulting in separated segments. This behavior is consistent with covalently bonded crystals such as CNTs and graphene. Note here that some simulations using the reax force field have reported a ductile deformation mode in nanocarbon materials [43]; this is different from the approach used in the present study. By computing the slopes of the stress–strain curves, our results reveal that armchair PNTs possess a higher Young’s modulus than zigzag PNTs. The average value for armchair PNTs is roughly 214 GPa, while it is approximately 154 GPa for zigzag PNTs (see Appendix A). We attribute this difference to the alignment of the carbon chains. In armchair PNTs, the chains are aligned with the PNT axis, whereas in zigzag PNTs, they form angles with the axis, making them less capable of resisting tensile loading. Notably, the stress–strain plot for the 11.91 Å zigzag PNT exhibits a much lower slope. As a result, its modulus is much smaller (around 114 GPa) than that of the larger-sized zigzag PNTs (160–170 GPa). Upon analyzing the structure, we found that for zigzag PNTs, the length contracts significantly with a decreasing diameter, similar to the bending Poisson effect observed by Liu et al. [44]. (Here the contract ratio is measured by **(*L_*0*_ − L*)/*L_*0*_*, where *L*_0_ is the length of the original model after the rolling up of the defected graphene sheet, and *L* is the length after geometry optimization). Consequently, when subjected to tensile loading, the response of the PNT is slower. Although armchair PNTs also exhibit longitudinal contraction, the contraction ratio is not sensitive to their diameters (see Appendix A), resulting in a weaker diameter dependence of the Young’s modulus.

Comparing PNTs with CNTs and graphene, it can be concluded that the ultimate stress and Young’s modulus of PNTs are significantly lower, approximately 30 GPa and 184 GPa, respectively. In contrast, the corresponding values for CNTs are approximately 200 GPa and 1 TPa. This is attributable to the fact that PNTs fall under the category of defective CNTs, impeding their capacity to carry external loads. The overall mechanical characters investigated in this study are summarized in Table 1 and Table 2.

Figure 5c,d display the extracted ultimate stress and strain from the stress–strain curves, revealing that the aspect ratio has no considerable impact on the ultimate stress and strain of the PNTs. The diameter, on the other hand, plays a crucial role. The ultimate strain demonstrates a decreasing trend with an increasing diameter until it converges, with the ultimate stress displaying similar behavior. This reason is due to two aspects: firstly, as discussed earlier, PNTs contract at small diameters, facilitating deformation under tension; secondly, carbon chain twisting sensitivity varies according to the tube diameter. For small-diametered PNTs, the degree of twisting is stronger (up to 80°), leading to the untwisting of carbon chains under tensile loading. Figure 6 represents a typical result obtained from a D = 16.50 Å PNT. The twist angle steadily drops from 80° to 55° as the tensile strain rises to 12%. It then remains relatively stable before the breaking point when a failure transpires at a strain level of 21%, demonstrating an almost complete fracture of the PNTs since the twist angles fluctuate and return to values near the original ones. The immediately fractured structures of both armchair and zigzag PNTs are shown in Figure 7, where brittle failure mode is clearly seen.

Figure 8 illustrates the compressive behavior of PNTs featuring various chiralities and diameters. It is noteworthy that the D = 16.5 Å PNT exhibits a substantially higher stress level than the other PNTs studied in our studies, showing a peak stress value of 9.4 GPa. In contrast, the peak values for other armchair PNTs are lower than 3 GPa. This unusual behavior arises from the progressive collapse of cross-sectional geometries, meaning that larger diametered PNTs are less capable of withstanding compressive loadings. On the other hand, for zigzag PNTs, the critical stress shows a slightly different trend regarding the PNT diameter, i.e., it gradually decreases with the diameter. As discussed earlier, this is partly due to the relatively well-preserved circular cross-sectional geometry in zigzag PNTs. Additionally, as the linear carbon chain does not align along the axial direction, the maximum compressive stress that zigzag PNTs can withstand is smaller than that of armchair PNTs. The inverse correlation between PNT diameter and the critical buckling stress aligns with previous CNT buckling theories [45]. 

Interestingly, the critical buckling strain for zigzag PNTs is substantially higher than that for armchair PNTs. The underlying mechanisms can be better comprehended by investigating the structural evolution of two representative PNTs, as illustrated in Figure 9. Upon observation, it becomes evident that for the armchair PNT, the compressive stress is primarily experienced by the linear carbon chains along the axial direction, rapidly reaching a critical buckling strain similar to that in a rod. Meanwhile, for zigzag PNTs, the load transfer pathway is not along the axial direction, and the deformation mode in the linear carbon chains is not exclusively compression but is also bending-induced bond angle change. Therefore, the strain energy stored in the zigzag PNT is significantly smaller at the same level of strain, thereby leading to delayed buckling events. This scenario is further corroborated by energy storage density simulations performed in our experiments, as shown in Figure 10. Here, the energy storage capacity is calculated as the following:(1)ΔEM=ΔEn(C)×Ar(C)×m(C12)12+n(H)×Ar(H)×m(C12)12
where ΔE represents the energy change in the relaxation process; n represents the number of atoms; $A_{r}$ represents the relative atomic mass; m(C12) represents the carbon-12 mass. The average energy density for armchair PNTs is approximately 30,000 KJ/Kg, while for zigzag PNTs, it is about 19,200 KJ/Kg. While an energy density lower than CNTs is expected, the density is nevertheless higher than recently predicted values of nanothread [46].

## 4. Conclusions

Extensive molecular dynamics simulation studies have been conducted on the structural and mechanical properties of recently synthesized phenine nanotubes (PNTs). The results demonstrate that due to distortion in their linear carbon chains, PNTs exhibit distinct cross-sectional geometry variation trends depending on their chirality and diameters. These phenomena further modify the PNTs’ mechanical responses to tensile and compressive loadings. Armchair PNTs are stronger than zigzag PNTs, as evidenced by a higher Young’s modulus and energy storage capacity, which can be attributed to the alignment of the linear carbon chains along the axial direction. However, PNTs’ overall mechanical properties are weaker than those of carbon nanotubes (CNTs), since they can be regarded as defective CNTs. The intriguing role of twisting-untwisting in tuning the mechanical properties of PNTs offers new insights into the mechanics of advanced nanostructures. It is expected that these results will inspire future studies aimed at designing new functional materials with desirable properties.

## Figures and Tables

**Figure 1 materials-16-04706-f001:**
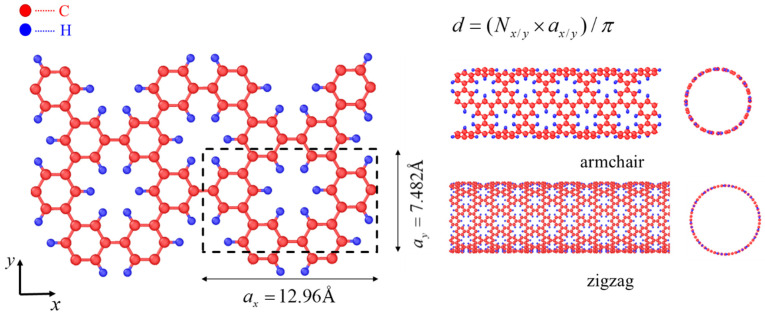
Rolling up of porous graphene (left) to phenine nanotubes (right). On the left, a six-membered ring of carbon in the unit cell of a graphene sheet is replaced by 6 hydrogen atoms, forming a periodically defected structure; the size of the unit cell is 12.96 Å × 7.48 Å. Once seamlessly rolled up along armchair or zigzag edges, armchair PNTs or zigzag PNTs are formed, and the diameter of the tubular structure can be simply estimated.

**Figure 2 materials-16-04706-f002:**
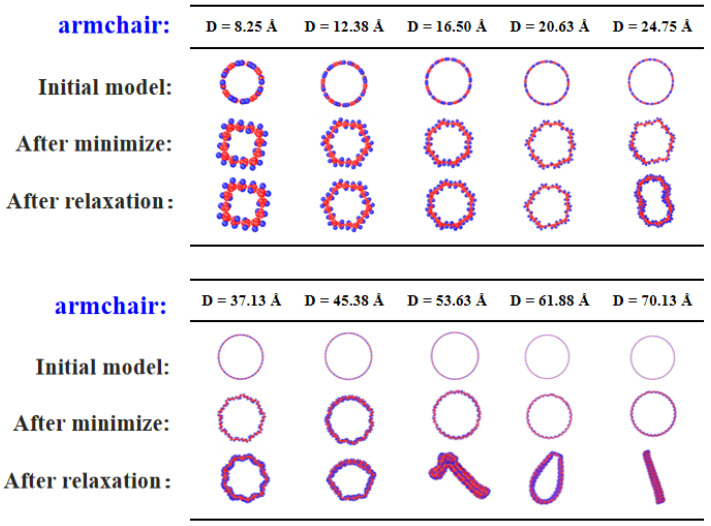
Evolution of cross-sectional geometry of armchair PNTs with diameter (ranging from 8.25 Å to 70.13 Å). For each case, the initially constructed atomic model, the structure after energy minimization, and those after structural relaxation at 1 K are provided.

**Figure 3 materials-16-04706-f003:**
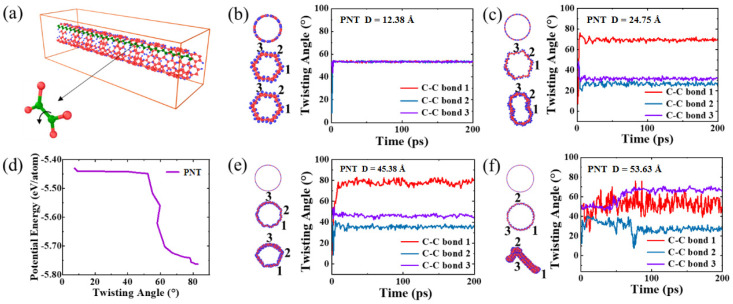
Twisting of zigzag carbon chains in armchair PNTs (**a**). The potential energy of the system decreases as a result of twisting (**d**). (**b**,**c**,**e**,**f**) present selected twisting angle evolution as a function of time for PNTs with different diameters.

**Figure 4 materials-16-04706-f004:**
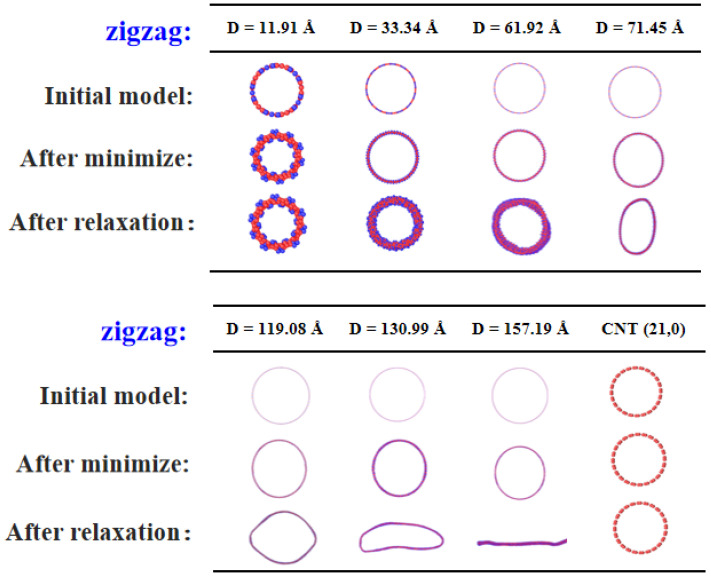
Evolution of cross-sectional geometry of zigzag PNTs with varying diameter (ranging from 8.25 Å to 157.19 Å). For each case, the initially constructed atomic model, the structure after energy minimization, and those after structural relaxation at 1 K are provided. The structures of a finite sized zigzag CNT are also provided on the lower right panel for comparison.

**Figure 5 materials-16-04706-f005:**
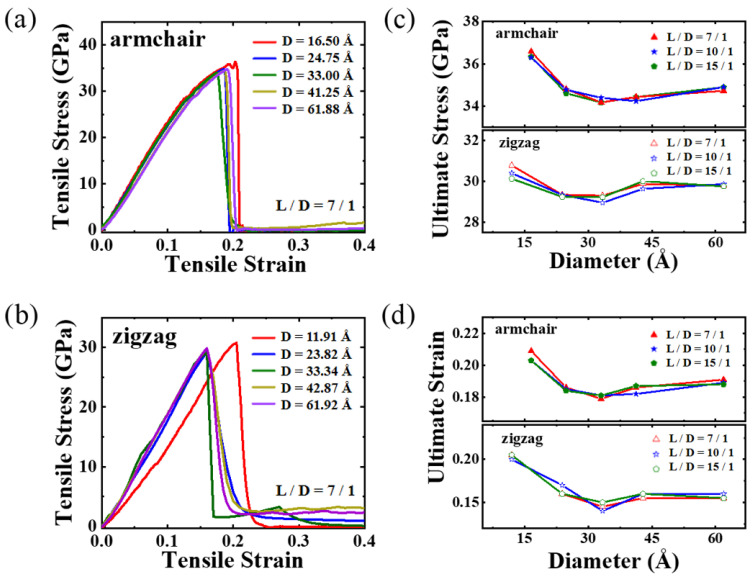
Mechanical properties of PNTs under tensile loadings. (**a**,**b**) show stress–strain relationship for armchair and zigzag PNTs with aspect ratio of 7/1, the ultimate stress and ultimate strain are plotted in (**c**,**d**) as a function of diameter.

**Figure 6 materials-16-04706-f006:**
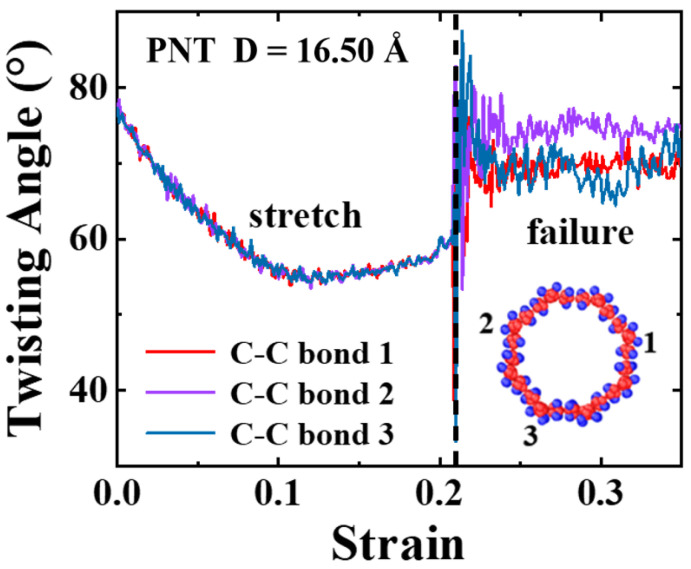
Evolution of twisting angle for a small diametered PNT under tensile loadings. Three chains are selected for the calculation; the locations are indicated in the inset.

**Figure 7 materials-16-04706-f007:**
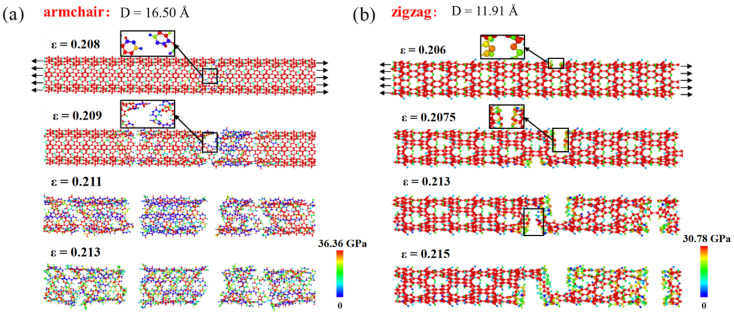
Failure process of PNTs under tensile loadings. The stress distribution for each PNT is provided and the stress level can be referred to the color bars. (**a**) armchair PNTs; (**b**) zigzag PNTs.

**Figure 8 materials-16-04706-f008:**
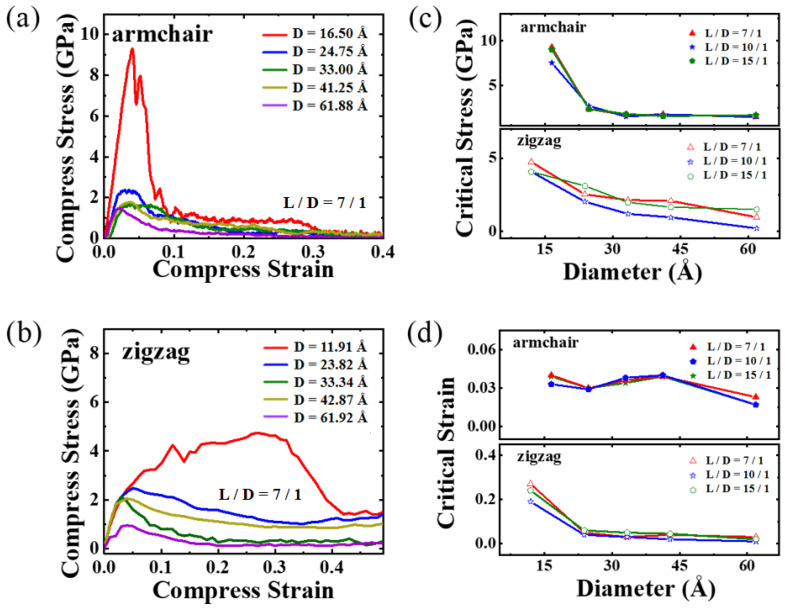
PNTs under compressive loadings. (**a**,**b**) show the stress–strain relationship for the armchair and zigzag PNTs. The critical stress and critical strain for buckling are shown in (**c**,**d**).

**Figure 9 materials-16-04706-f009:**
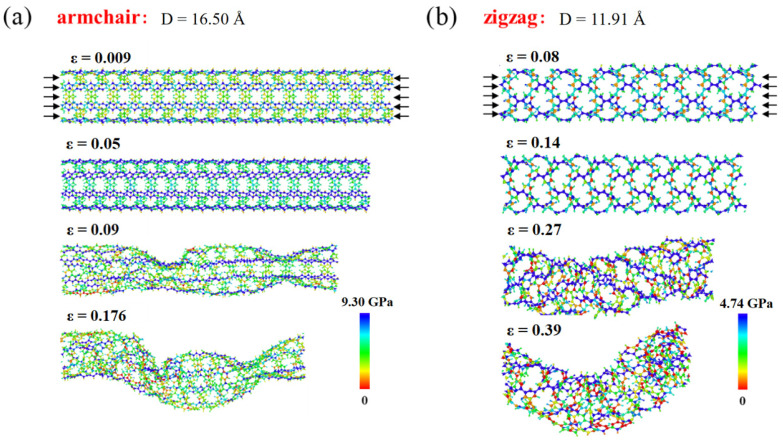
Buckling process of PNTs under compression. The stress distribution for each PNT is provided and the stress level can be seen from the color bars. (**a**) buckling process of an armchair PNT with D = 16.5 Å, (**b**) buckling process of a zigzag PNT with D = 11.91 Å.

**Figure 10 materials-16-04706-f010:**
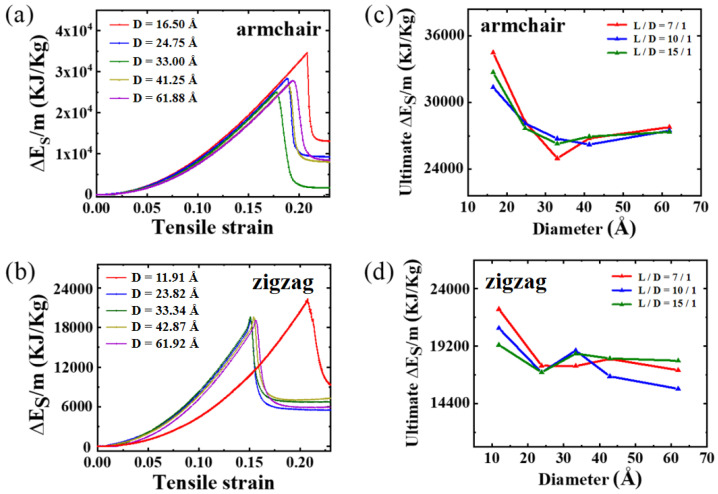
Energy storage density of PNTs under tensile loadings.

**Table 1 materials-16-04706-t001:** Summarized mechanical properties of armchair PNTs.

Diameter	16.50 Å	24.75 Å	33.00 Å	41.25 Å	61.88 Å	Unit
Ultimate stress under tension	36.36	34.80	34.17	34.43	34.72	GPa
Ultimate strain under tension	0.209	0.186	0.179	0.186	0.191	-
Young’s modulus	239	219	210	199	197	GPa
Energy storage density	34,488	28,248	24,984	26,784	27,792	KJ/Kg
Critical stress under compression	9.30	2.38	1.69	1.78	1.48	GPa
Critical strain under compression	0.04	0.03	0.036	0.039	0.023	-

**Table 2 materials-16-04706-t002:** Summarized mechanical properties of zigzag PNTs.

Diameter	11.91 Å	23.82 Å	33.34 Å	42.87 Å	61.92 Å	Unit
Ultimate stress under tension	30.78	29.34	29.30	29.87	29.79	GPa
Ultimate strain under tension	0.205	0.16	0.145	0.155	0.155	-
Young’s modulus	114	160	173	166	159	GPa
Energy storage density	22,296	17,568	17,544	18,132	17,196	KJ/Kg
Critical stress under compression	4.74	2.52	2.13	2.07	0.95	GPa
Critical strain under compression	0.27	0.05	0.03	0.04	0.03	-

## Data Availability

Available upon request to corresponding authors.

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
