# Peer review of "Size- and Chirality-Dependent Structural and Mechanical Properties of Single-Walled Phenine Nanotubes"

_materials, 2023, doi:10.3390/ma16134706_

Round 1
Reviewer 1 Report
The manuscript by Y. Liu et al. addresses the influence of chirality, length, and diameter of phenine nanotubes on their mechanical behavior. The authors subjected different size phenine nanotubes to compression and traction. First, the paper should not be published without a proper language revision, since this is, in my view its most serious flaw. On the other hand, the originality and importance of the results presented here are, at best, medium. The results are new, but given other works published on this subject, there is nothing novel here, neither in the method, nor in the challenges addressed. The paper is concise and more or less well organized. However, I would support the publication of this paper if the authors address the following observations:
a) In the simulation details section, the OVITO software should be referenced.
b) How were the different PNTs constructed? What software did you use?
c) Why did you use a timestep of 1 fs, why not longer? How was the stretching/compressing done, by applying fix deform to the simulation box? Why use this strain rate, as it been used in previous works? Did you use the same strain rate both in compression an traction? This section lacks important details!
d) In figure 1, zigzag PNT is not clearly represented, please improve it.
e) The introduction of a CNT representation, without mentioning the chirality or diameter, in Figure 3, when it was not also introduced in Figure 2, should be avoided since it is not useful. Figures 2 and 3 were divided in two rows, but the titles Initial model; After minimize and After relaxation only affect the upper row, when should affect both.
f) After Figure 3, you claim that “It is easy to observe from the stress-strain relationship that both type of PNTs are brittle materials as stress drops immediately to zero after reaching their peak values and the structures break into separated segments, this is common for covalently bonded crystal structures such as CNTs and graphene” In fact this is not entirely correct, but depends on the potential used and on the diameter, chirality and length of the “covalently bonded crystal structure”. It is certainly true for AIREBO with C-C cutoff minimum distance increased from 1.7 to 2.0 A. Necking and slow tearing apart of the graphenic mesh can be observed with Reax for example. You should mention this.
g) The authors should compare their results for the mechanical properties with for example the results from the paper by Faria and Silvestre, already referenced in this paper.
h) The authors should summarize their results for the mechanical properties in one or more Tables.
i) Please explain how you measured the nanotube contraction mentioned in “By examining the structure characters we found that for zigzag PNTs, the length of the structures contract significantly with decreased diameter, similar to the bending poisson effect observed by Liu et al “
j) Please clarify the significance of the following phrase “The reason comes from two aspects, the first aspect is the above mentioned contraction of PNTs at small diameters, therefore offers more room for deformation under tension “.
k) How did you measured the twist angles? It is important to clarify this point with schemes or visual representations so that the readers understand this phenomenon.
l) In Figure 5, the pNT in the legend should be PNT.
m) The authors have not identified the bond types in the main text but only in Figure S1. However they mention them in the main text, forcing the readers to read the supplementary material.
n) Please use pictures to help define bond twisting, it is very hard to understand otherwise.
o) How did you obtain the energy storage density. Explain why do some energy storage density curves do not begin a null strain, and why do they begin with an unnatural sudden sharp increase for very small strains.
Quality of English Language is low and should be revised
Reviewer 2 Report
see file

Author Response
Reviewer #2:
Comment 1: Does the chiral matter in this type of nanotubes? Or some reason why it wasn't parsed?
Reply: The chirality parameter does affect both the geometry and mechanical properties of the PNTs, different geometrical characters are discussed in Fig.2 and Fig.3; while the mechanical properties are also compared in the following discussions.
Comment 2: I suggest attaching important reference: Physica E 134: 114874 (2021)
Reply: Thanks for the suggestion, this reference is now cited.
Comment 3: Were the tubular structures analyzed in neutral charge? And does the introduction of point defects modify the multiplicity of systems? Was this situation analyzed? Magnetism could be present, for example.
Reply: The reviewer raised interesting points for studying physical properties of the PNTs. While the present study mainly involves mechanical responses of PNTs under neutral charged state as the dangling bonds are saturated by hydrogen atoms, not polar groups. We therefore did not focus on discussing of physical or chemical responses of the PNTs to external fields. It would be interesting to study how the zigzag edges of the structures responses to electric fields in future works as is widely known that zigzag edges of graphene nanoribbons exhibits magnetic ground state.
Comment 4: Because the authors have not made the vibrational calculations through DFT theory for example.
Reply: The vibration behavior is not considered in this work and its structural stability has been verified by experimental results, we therefore mainly focused on MD studies of its mechanical properties. DFT calculations on its properties have been reported by other groups, such as: Zhe Sun et al. theoretically investigated electronic structures of infinite pNT by extrapolating the (12,12)-pNT molecule, and obtained the electronic density of states (DOS) of infinite (12,12)-pNT [2]. Anton J. Stasyuk et al. studied electronic properties of pNT⊃C70 and modeled photoinduced electron transfer processes in the complexes using the TD-DFT theory [3]. Koki Ikemoto et al. confirmed optical properties of (12,12)-NpNT in the density functional theory (DFT) calculations [4]. Olga A. Stasyuk et al. studied photoinduced electron transfer in a series of inclusion complexes of structurally modified phenine nanotubes (pNT) with C70 using the TD-DFT method [5].
- Sun, Z.; Ikemoto, K.; Fukunaga, T. M.; Koretsune, T., Finite phenine nanotubes with periodic vacancy defects. Science 2019 363, 151-155.
- Stasyuk, A. J.; Stasyuk, O. A.; Sola, M.; Voityuk, A. A., Photoinduced electron transfer in nanotube supersetC(70) inclusion complexes: phenine vs. nanographene nanotubes. Chem. Comm. 2020 56, 12624-12627.
- Ikemoto, K.; Yang, S.; Naito, H.; Kotani, M.; Sato, S.; Isobe, H., A nitrogen-doped nanotube molecule with atom vacancy defects. Nat. Commun. 2020 11, 1807.
- Stasyuk, O. A.; Stasyuk, A. J.; Sola, M.; Voityuk, A. A., How Do Defects in Carbon Nanostructures Regulate the Photoinduced Electron Transfer Processes? The Case of Phenine Nanotubes. Chem. Phys. Chem. 2021 22, 1178-1186.
Comment 5: How does the polarity and cohesion of energy vary?
Reply: This comment is also about the charged states of the PNTs under the circumstance of functionalized by chemical groups, it is a very interesting topic, we would expect future DFT studies to clarify such issues. Thanks again for the fruitful suggestions.
Round 2
Reviewer 1 Report
The authors addressed in detail the observations made by the reviewers and modified the manuscript accordingly. Therefore I support the publication of this paper in the present form.